# Ultrastructure of a Mechanoreceptor of the Trichoid Sensilla of the Insect *Nabis rugosus*: Stimulus-Transmitting and Bio-Sensory Architecture

**DOI:** 10.3390/bioengineering10010097

**Published:** 2023-01-11

**Authors:** Shashikanth Chakilam, Rimvydas Gaidys, Jolanta Brożek

**Affiliations:** 1Department of Mechanical Engineering, Kaunas University of Technology, LT-51424 Kaunas, Lithuania; 2Faculty of Natural Sciences, Institute of Biology, Biotechnology and Environmental Protection, The University of Silesia in Katowice, Bankowa 9, 40-007 Katowice, Poland

**Keywords:** *Nabis rugosus*, trichoid sensillum, morphology, stimulus transmitting, microtubules, point of rotation/pivot point, sensory architecture

## Abstract

This paper presents the ultrastructure morphology of *Nabis rugosus* trichoid sensilla using SEM and TEM data, along with a two-dimensional model of the trichoid sensilla developed in Amira software. The SEM images show the shape and scattering of the trichoid mechanosensilla over the *N. rugosus* flagellomere. The TEM images present the ultrastructural components, in which the hair rises from the socket via the joint membrane. The dendrite sheath is connected at the base of the hair shaft, surrounded by the lymph space and the socket septum. This dendrite sheath contains a tubular body with microtubules separated by the membrane (M) and granules (Gs). This study presents a model and simulation of the trichoid sensilla sensing mechanism, in which the hair deflects due to the application of external loading above it and presses the dendrite sheath attached to the hair base. The dendrite sheath is displaced by the applied force, transforming the transversal loading into a longitudinal deformation of the microtubules. Due to this longitudinal deformation, electric potential develops in the microtubule’s core, and information is delivered to the brain through the axon. The sensilla’s pivot point or point of rotation is presented, along with the relationship between the hair shaft length, the pivot point, and the electric potential distribution in the microtubules. This study’s results can be used to develop ultra-sensitive, bioinspired sensors based on these ultrastructural components and their biomechanical studies.

## 1. Introduction

*Nabis rugosus* (Linnaeus, 1758) is one of the species in the Nabidae (Heteroptera: Cimicomorpha) family [1]. It is a great contributor to nature and agricultural pests’ natural predators [2]. Heteropterans have different types of sensilla on their body. These can efficiently sense the ecological changes in the parameters of the physical environmental [3,4,5].

The antennae of insects contain various sensilla-like chemoreceptors, mechanoreceptors, and thermo-hygroreceptors. They play vital roles in host recognition, aggregation, and protection, typically mediated by mechanical disturbances and chemicals produced by conspecifics and host plants [6]. Mechanoreceptors are abundant in arthropods and vary significantly in their structures and functions. With regard to mechanical properties, the trichoid sensilla of insects have drawn the most interest. These sensilla have drawn interest due to their extraordinary sensitivity, allowing them to be triggered by even the slightest mechanical stimuli. Mathematical simulations and experimental studies have been undertaken on the core biomechanics [7,8,9,10].

There are limited studies on the sensory connections and stimulation mechanisms of arthropods’ tactile hairs. In trichoid filiform hairs, the hair shaft is assumed to be a rigid structure that primarily undergoes deflection as a result of external forces, with these external or stimulating forces being due to close contact with an object [11]. Formulating and analyzing this process is difficult due to the geometrical asymmetries. The deformation of the hair shaft due to the stimulation determines the load applied above the hair [12]. The magnitude of these hair deformations by mechanical stimuli leads to geometrical nonlinearities, which cannot be ignored or simplified with linear theory, as an undeformed system is not valid [13]. To develop a numerical model of trichoid sensilla, the following analysis discretized the complex problem. For this analysis, the hair morphology and its material properties were determined. It was also important to measure tiny forces (in the order of micronewtons), which could be used to accurately quantify hair deflection and deformation [14]. These processes and data forms were the basis of the present study. The novelty of trichoid sensilla stimulus–response can be revealed by determining their morphology and biomechanics.

In this study, the ultrastructure of *N. rugosus* trichoid sensilla was studied in detail with the help of SEM and TEM. The connection between the sensilla hair base and the dendrite is crucial for the stimulation and transmission process.

A two-dimensional model was developed based on a sectional view of the trichoid sensilla obtained using Amira software from TEM data. The numerical model was developed using COMSOL software, and structural mechanics and electrostatics studies were used to determine the dendrite sheath displacements and the microtubules’ generation of electric potential.

## 2. Materials and Methods

### 2.1. Specimen Collection

*Nabis rugosus* specimens were procured from Silesian Park, Katowice, Poland, and preserved in 80% ethanol for SEM studies; for TEM studies, these specimens were conserved in 2.5% glutaraldehyde. The species was identified using the key for damsel bugs [15].

#### 2.1.1. Samples under Scanning Electron Microscope

*Nabis rugosus* specimens preserved in ethanol were used. The antennae were detached from the head with scalpels using an Olympus stereomicroscope (SZX7) and washed using an ultrasound cleaner (Polsonic, Warsaw, Poland) for about a minute. The antennae were dehydrated using ethanol solutions ranging from 80% to 90% for 15 min, bathed twice with absolute ethanol for about 10 min, and then dried at room temperature. These specimens were placed over the carbon tapes and, to enhance the conductivity of these materials, the surfaces were coated with gold spray sputter (20 nm) using a Quorum 150T ES Plus (Quorum Technologies, Laughton, East Sussex, UK). At the Faculty of Natural Sciences of the University of Silesia in Katowice, images of these specimens were captured using a Hitachi UHR FE-SEM SU 8010 (High Technologies, Tokyo, Japan) and Phenom XL SEM (Phenom-World, the Netherlands) [16].

#### 2.1.2. Samples under Transmission Electron Microscope

*Nabis rugosus* were numbed and then flagellum portions of the antenna were cut and placed in 2.5% C_5_H_8_O_2_ solution at 4 °C for about 24 h. After fixation in glutaraldehyde solution, they were cleaned with phosphate buffer three times (thirty minutes each at room temperature) and then in 2% O_s_O_4_ for two hours at room temperature. Again, they were cleaned with phosphate buffer (10 min each at room temperature three times). The material was dehydrated by using different concentrations of ethanol (with 50%, 70%, 90%, and 96% for about 10 min each and with absolute ethanol four times for about 15 min each) and with the solution of (1:1) 100% C_2_H_6_O and C_3_H_6_O for about 15 min and twice with acetone for about 15 min. These materials were treated with C_3_H_6_O and epoxy resin for about 1.5 h. Epoxy resin (Epoxy Embedding Medium Kit, Sigma, St. Louis, MO, USA) is used to embed these materials. Using Leica EM UC7 RT ultramicrotome (Frankfurt, Germany), the material was cut into ultrathin sections (50 nm each). A TEM (Hitachi H500, Tokyo, Japan) operating at 75 kV was used to analyze ultrathin sections placed over copper grids. Ninety-six sectional images of Trichoid sensilla were acquired serially at 50 nm intervals. All images were captured with 1024 × 1024 pixel resolution with an effective magnification of ×18 and were saved as TIFF files. These were carried out at the Faculty of Natural Science, TEM laboratory of the University of Silesia in Katowice [17,18].

#### 2.1.3. Image Processing—AMIRA

MIB software (Microscopy image browser software) (University of Helsinki, Finland) was used to stack the images produced by TEM from ultrathin sections. They are aligned and cropped, then stored with the *. Am file. Thermo Fisher Scientific’s Amira 6.5 software (Amira 3D2021.2, Thermo Fisher Scientific, Waltham, MA, USA) generates the surfaces and volume rendering for these microscopic images [19]. The generated files are edited using Adobe Photoshop CS6 and CorelDraw Graphics suite 2021.

### 2.2. Theoretical Model of the Hair Mechanoreceptor Trichoid Sensilla

#### 2.2.1. Mathematical Model

Researchers have developed several models to analyze the relationship between the trichoid sensilla and their responsiveness to various stimulations. Kanou et al. [20], Kumagai et al. [21], and Shimozawa et al. [22] proposed that trichoid sensilla were assumed to be an inverted pendulum model with a torsional spring and torsional damper elements, which counters the angular movement of the hair shaft, as shown in Figure 1.

The equation of the hair shaft motion is assumed as
(1)$$I\frac{d^{2}\theta }{{\mathrm{dt}}^{2}}+R\frac{d\theta }{\mathrm{dt}}+S\theta =N$$
where I represents the moment of inertia with regard to the point of rotation, R represents the torsional damping constant at the hair base, S represents the hair base restoring constant, θ is the angular displacement of the hair base $\frac{d^{2}\theta }{dt^{2}}\;\mathrm{is}\;\mathrm{the}\;\mathrm{angular}\;\mathrm{acceleration}\;\mathrm{with}\;\mathrm{regard}\;\mathrm{to}\;\mathrm{time}\;\left(t\right)$, and $\frac{d\theta }{dt}\;\mathrm{is}\;\mathrm{the}\;\mathrm{angular}\;\mathrm{velocity}\;\mathrm{with}\;\mathrm{regard}\;\mathrm{to}\;\mathrm{time}\;\left(t\right).\;$The torque N balances the sum of the angular momentum of any form of motion and at any time instance, where I is the moment of inertia of the hair, R is the viscous resistance within the hair base, S is the spring stiffness, and N is the torque applied by the contact forces.
(2)$$N=\int_{0}^{L}f_{y}y\;\mathrm{dy}$$
where f_y_—the force per unit length, y—distance variable, and L—the hair length.

Torque N is produced due to the external force applied on the hair shaft at a distance y from its base (Figure 1). The hair shaft rotates about its base due to the developed torque N being equal to the integration of all torques generated at infinitesimal elements of hair-shaft lengths L [21].

The moment of inertia of the hair about its base is given by the integrated sum of the second-order moment of mass distribution along the length of the rotational arm
(3)$$I=\int_{0}^{L}y^{2}\mathrm{dM}$$
(4)$$\mathrm{dM}={\rho \pi r}^{2}\mathrm{dy}$$
where dM is the mass of the hair shaft, L is the length of the hair shaft, ρ is the hair-shaft density, and r is the radius of the hair shaft.

The maximum deflection at the hair-shaft tip is given by
(5)$$\delta_{\max}=\frac{f_{d}L^{3}}{3\mathrm{EI}}$$

The maximum slope or the deflection angle at the hair-shaft tip is given by
(6)$$\theta_{\max}=\frac{f_{d}L^{2}}{2\mathrm{EI}}$$
where E is the young modulus, L is the length of the hair shaft, I is the moment of inertia of the hair shaft, f_d_ is distal force.

The insect sensilla architecture is considered as a first-order lever rigid body [22]. Hair sensilla are in a simple lever form that allows for any external contact forces by air or water over them, which compress the dendrite attached at the basal part. As we know the pivot point or point of rotation, the ratio of hair-shaft deflection at the tip to base is equal to the ratio of lengths of the hair shaft below and above the pivot point.
(7)$$\frac{\delta_{\mathrm{Base}}}{\delta_{\mathrm{Tipmax}}}=\frac{L_{1}}{L_{2}}$$

We also know that the ratio of the distal force applied on the hair shaft to the basal force of the hair shaft is equal to the ratio of lengths of the hair shaft below the pivot point and above the pivot point [23].
(8)$$\frac{f_{d}}{f_{b}}=\frac{L_{1}}{L_{2}}$$

#### 2.2.2. Geometrical Model and Numerical Analysis

A 2D geometric model of the trichoid sensilla ultrastructure was developed using AMIRA software from the cross-sectional images obtained from TEM. To be more precise of the entire model, as per the microscopic images, the socket was employed with the socket septum and lymph cavity around the dendritic path. A numerical model was created for multiphysics analysis of the trichoid sensilla by COMSOL multiphysics. The 2D space dimensional structural mechanics physical interface was used for the stationary study of the interaction between the hair and the dendrite, along with other components of the sensilla when there is an external excitation on the hair. The analysis was conducted in the piezoelectric-effect multiphysics with the stationary study.

### 2.3. Trichoid Sensilla Mechanical Properties

The sensilla mechanical properties have not been accurately presented or published by any researcher. Previous researchers assumed all these values, which were taken for analysis [24,25,26,27,28,29,30] and are presented in Table 1. The hair and socket were modelled as a linear elastic material. Microtubules considered are of piezoelectric components (PZT-5h) [24].

## 3. Results

### 3.1. Trichoid Sensilla Ultrastructure

Trichoid sensilla external morphology was identified on the antennae of *N. rugosus* under a scanning electron microscope in Figure 2. They were the most-abundant sensilla on the second flagellomere of *N. rugosus* antennae. Trichoid sensilla Ts are long, straight or curved hairs with a pointed tip, with their bases tightly inserted into the cuticle through sockets (Figure 2a). The hair shaft Hs stands perpendicular to the cuticular surface C and embedded into the socket and has a non-porous wall when it is viewed under high magnification of 1 nm at 1 kv (Figure 2b). Figure 2c shows that the hair shaft Hs connects to the socket S along with a joint membrane J, which is elastic in nature.

The ultrathin structure along the longitudinal section is documented and thoroughly described in Figure 3. The joint membrane (J), the suspension fibers (Sf), and the socket septum (Ss) are distinctly seen by the partition of the socket structure (S) in Figure 3a.

In Figure 3a, we observed that these trichoid sensilla had a single bipolar sensory cell, in contrast with other receptors (in some cases, chemoreceptors have four sensory cells). The dendrite, surrounded by a dendrite sheath (Ds), connects to one side of the hair-shaft base via the lymph space (L). This material is continuous with the socket septum (Ss), which surrounds the dendritic sheath. This socket septum (Ss) brimmed up to the socket layers (S).

The *Nabis rugosus* tubular body structure was studied and is presented in the present study, where its diameter is about 0.15 µm. It resembles a cylinder with a flat side facing (Figure 3a,b). The tubular body comprises approximately 90 microtubules (Figure 3b). Inside this tubular body, not even a single neurofilament was identified. Microtubules of this tubular body are not bound to the dendrite sheath, as they are partitioned with the membrane M and Granules G. (Figure 3b). Based on the compilation of numerous thin slides from longitudinal sections near the base of the hair, we analyzed the components of the tubular body and presented them in the structure model.

A reconstructed 2D model of trichoid sensilla sectional view using Amira software is shown in Figure 4, and the components are hair shaft, joint membrane, socket, suspension fibers, socket septum, lymph space, and epithelial layer. The dimensions of the sensilla of the *Nabis rugosus* were derived from Amira software. The hair length is 21.66 µm, the length of the joint membrane is 10.76 µm, and the depth is about 4.49 µm; the socket diameter at the cuticle part is about 15.70 µm, and at the base, it is about 17.37 µm. The socket septum, which is attached from the socket and was rounded to the dendrite sheath, is about 11.26 µm length, and the dendrite’s tip is about 0.54 µm and has a 3.07 µm diameter in its middle.

### 3.2. Computational Scheme of the Trichoid Sensilla—External Excitation and Boundary Conditions

The boundary conditions for the computational analysis are fixed outer edges of the socket. The cuticle is rigid in nature and is firmly fixed to the insect’s body, and the socket is embedded into the cuticle, thus, being fixed. A point loading of 1 µN force was applied over the hair, as shown in Figure 5a. At first, the displacements in the dendrite due to the hair deflection were investigated. The pivot point of the hair shaft and the dendrite connection due to the deformation was investigated. According to previous researchers, the rotation point should be between 1 µm and 2 µm from the hair-shaft base and near the connection site [31,32].

To evaluate the electrical potential in the microtubules due to the hair-shaft deflection, a loading of 1 µN force was applied on the hair shaft, considered four microtubules in length 5.76 µm each and the width of each microtubule is 0.17 µm and was grounded at the apex of the microtubule near the dendrite sheath connection, fixed at the base, as shown in Figure 5b.

### 3.3. Bio-Micro-Electro-Mechanical System of the Hair-Type Mechanoreceptor Trichoid Sensilla

This study examines the effect of hair deformation when subjected to loading. The results found that the hair-shaft deflection is due to an external force of 1 µN over the hair shaft at its tip in the distal end. The hair deflects due to the loading along the loading direction, and the base of the hair shaft presses the dendrite sheath at the connection site. The dendrite sheath is displaced against the loading direction, as shown in Figure 6. The hair shaft deflected at an angle of 6.68 deg along the applied force (the deflection angle ranges from 6 to 8 deg for the application of 1 µN, already described by a previous research paper [33]) and the point of rotation is at 1.45 µm from the hair base (as described by Keil [32], it will be around 1–2 µm from the base of the hair shaft and near to the point of the connection of the hair base and dendrite sheath) are shown in Figure 6a. The distal hair-tip deflection angle is maximum when compared to the deflection angle at the basal hair [33]. Figure 6b shows the distal hair-shaft deflection angle, which is about 6.68 deg, and the basal hair-shaft deflection angle is about 0.48 deg. In turn, the basal hair shaft, which is in connection with the dendrite sheath tip at the connection site, moves in the opposite direction to that of the distal hair shaft and perpendicular to the microtubules in the tubular body of the dendrite sheath at about 4 nm in the horizontal direction, shown in Figure 6c. The dendrite sheath compression along the vertical direction is shown in Figure 6d.

In this study, a computational method for an electro-mechanical model for microtubules was developed. The load applied at the tip of the hair shaft displaced the dendrite sheath at the connection of the hair base, and the longitudinal deformation of the microtubules takes place under the application of transversal deformation from the dendrite sheath, as shown in Figure 7a, as the compression of the microtubules taking place at the point of connection with the dendrite sheath.

From Figure 7b, we observe the different electric potential generations in the core of each microtubule. Microtubules near to the connection site have maximum electric potential and microtubules away from the connection site have lower electric potential at the base of the microtubule, which stimulates the axon and the brain of the insect. The distribution of electric potential in the core of the individual microtubules is presented in Figure 8. It can be seen that the potential variation is not completely linear.

## 4. Discussion

### 4.1. Trichoid Mechanosensilla Morphological Parameters

In *N. rugosus*, trichoid mechanosensilla were found abundantly on the pedicel and flagellomeres of the antennae. On these antennae, the sensilla are spread in different patterns and vary in size. *N. rugosus* sockets are similar to a Cricket insect’s *(Acheta domesticus*) filiform hair sockets, as in Figure 2c [34]. *N. rugosus* trichoid sensilla (Figure 2b,c) are non-porous straight hairs, the same as the Cricket insect (*Gryllus bimaculatus*) filiform hairs [31,35].

Various researchers explained that the different insects’ mechanoreceptor microstructure and the organizational structure of trichoid sensilla are almost similar [32,36] but vary with the number of microtubules, which are significantly sensilla actions. The external morphology of the trichoid sensilla varies from other mechanosensilla, such as trichobothria and campaniform sensilla. The external sensilla emerge from the cuticular layers, which consist of the exo- and endocuticle.

In this study of the trichoid sensillum of *N. rugosus*, a clear sectional view of the internal ultrastructure was observed (Figure 3a) as well as the compact structure of the dendrite sheath where microtubules are packed in EDM and are separated by membrane M and granules Gs from the dendrite sheath (Figure 3b).

The dendritic sheath (the “cuticular Scheide”) [31], which is developed from the thecogen cell, is found in all trichoid sensilla. The dendrite sheath is connected to the plasmic membranes through the cylindrical granules Gs. These granules are bridge-like structures and are very minute. One morphological characteristic of stimulus transmission over the tubular body is the dendrite sheath thickness [37]. For the functional morphology, the dendritic sheath only has contact with the tubular body at the distal edge of the dendrite connecting with the base of the hair shaft [38,39]. In the study of the dendrite of the trichoid sensilla of *N. rugosus*, the tubular body is very stable when compared with other arthropods, such as *Locusta*, not having a stable suspension tubular body at its distal end. Compared with other arthropods, in *N. rugosus* trichoid sensilla at the hair shaft base and the dendrite sheath connection site, the stimulus is effectively transmitted to the tubular body due to the hair-shaft deflection.

The tubular body is described as the “insect epithelial mechanoreceptors where the stimulus elicits a neural response” by Thurm [40]. Except in mechanoreceptors, arthropod’s other receptors cannot find similar tubular bodies due to the “modality-specific differentiation of cuticular mechanoreceptors” [41]. In many arthropod trichoid sensilla, these tubular bodies are parallelly organized with the microtubules and the neurofilaments. In contrast, in *N. rugosus*, neurofilament was not found, which is similar to the filiform hairs of *Gryllus*, whereas one neurofilament was found in the *Nilaparvata lugens*. Generally, microtubules are surrounded by EDS (electron-dense substance). In the *N. rugosus* trichoid sensilla tubular body (Figure 3b), a dark EDM/EDS substance was observed, the bridge material that connects microtubules to one another. In contrast, in other insects’ mechanoreceptors, such as *Gryllus*, all the microtubules are encased in EDS in a condensed form. These tubular bodies are of a cylindrical rod and can be deformed by hair-shaft deflection [42]. The microtubules connect to the cytoplasmic membrane with small membrane connectors known as granules, also called filaments, which transfer the load from the dendrite sheath to the microtubule core [43].

### 4.2. Hair Deflection and Mechanism of Stimulation Transmission

Trichoid sensilla hairs are the external structures. These are considered as first-order levers, where the maximum deflection of the hair shaft at its tip transforms into a small deflection at the base of the hair shaft. This basal hair-shaft deflection compresses the dendritic sheath’s tip at the hair base’s connection site [32].

The hair shaft has no direct contact with the socket boundaries and it is 5.21 µm from the upper circular opening to the cuticular layer of the socket from which it projects, as shown in Figure 6b. The sensilla have a directional sensitivity, as the hair-shaft deflection is constrained to a specific plane, which is perpendicular to the body surface [44]. The hair base is bilaterally symmetrical and the hair base is at the middle of the socket and is being supported with the help of a joint membrane and other linkages, such as the suspension fibers and lymph space (L) (Figure 3a). The joint membrane acts as an elastic spring, which helps in restoring the hair shaft to its original position after stimuli. The other components,, such as the socket septum, help hold the dendrite in a stable position by which the stimulation may be delivered more effectively and also helps in the protection of the dendrite tip. Stimulation occurs due to hair-shaft deflection by which the hair-shaft base rotates about its pivot point or point of rotation [45]. Therefore, the hair shaft can be considered as a first-order lever with a defined pivot point. From Figure 3a and Figure 4, the hair-shaft base has direct contact with the dendrite tip, whereas, in *N. lugens*, the dendrite tip will be at the center of the hair-shaft base with no direct contact [36]. The distance of the pivot point from its hair base varies from sensillum to sensillum but will be about 1–2 μm from the hair base [32]. In *N. rugosus*, the point of rotation is at 1.45 μm from its hair base and very close to the connection site of the hair base and dendrite sheath (Figure 6a). The length from the hair base to the pivot point can be considered as the short-lever-arm length, and the length above the pivot point or point of rotation can be considered as the long-lever-arm length. In *N. rugosus*, the sensilla hair lengths are about 20 μm to 30 μm, whereas, in *Drosophila*, the hair lengths are about 100 μm to 150 μm. In *N. rugosus*, bristles have more rigidity than the dendritic tip, as they are made up of myofibrils, which are “cross-linked actin filaments” [46]. The hair-shaft length below the pivot point or point of rotation (L1) is 1.45 µm, the hair-shaft length above the pivot point (L2) is 20.21 µm, and the length ratio of the hair shaft below the pivot point and above the pivot point (point of rotation) is about 0.071.

Considering that the hair shaft is a lever arm and the applied external load onto the hair shaft is 1 μN, the hair-shaft deflection angle at the distal end is 6.68 deg and the hair shaft deflection at the basal end is about 0.48 deg (Figure 6b). The force applied at the hair shaft at its tip increases the force developed at its base so that the distal signal is amplified at the transduction site, which is the connection of the hair shaft and the dendrite. [30]. From Equation (8), the applied load is 1 μN at the tip of the hair shaft, which increased the force at the hair base to 13.93 μN (around 14 μN), by which the dendrite sheath was displaced for about 4 nm against the application of the external force on to the hair shaft at its distal end (Figure 6c), and in *Calliphora* macrochaetes on the thorax, the length is about 25 μm, which is approximately the same compared to the length of *N. rugosus* sensilla, where the distal hair shaft deflects at 7 deg; therefore, the maximal displacement of the dendrite sheath is 3e−4 nm [47]. Compared with the *Calliphora*, the *N. rugosus* can be stimulated and generate potential in the tubular body, even with a very small applied force based on the maximum displacement of the dendrite sheath.

Deflection of the hair shaft from its resting position accompanies the sensing mechanism of the trichoid sensilla. When the hair shaft deflects, the dendrite simulation occurs as the hair base at the connection site with the dendrite sheath moves perpendicularly to the axis of the tubular body, which is tightly packed with microtubules, resulting in a longitudinal compression, as shown in Figure 7. These microtubules work in the suspension due to the force generated and the displacement of the dendrite sheath. The microtubules are piezoelectric in nature [48]. As these microtubules, which are attached to the dendrite sheath, undergo longitudinal compression, the electric potential develops within its core, as shown in Figure 7b. *Drosophila macrochaetes* has nearly 350 microtubules, which generate a maximum electric potential of about 2 microvolts [49,50], whereas *N. rugosus* has nearly 90 microtubules generating a maximum potential of approximately 1.19 microvolts in each microtubule. The electric potential developed in the core is different in different microtubules [24].

## 5. Conclusions

In this paper, SEM images of the *N. rugosus* antennal segment show the distribution and external shape of the Trichoid mechanosensilla. Trichoid mechanosensilla ultrastructural components are shown using TEM, in which the sensing plane passes through the hair base where the dendrite sheath connects. This study also discusses the structure and components of the tubular body. This sensillum comprises a hair shaft that is 22 µm long, the socket, joint membrane, suspensions fibers, dendrite sheath and the tubular body are 150 nm in diameter, with nearly 90 microtubules.

The numerical model was created and simulated using COMSOL multiphysics software. The point of rotation (pivot point) was obtained at 1.45 µm from the hair base. The length of the hair shaft below the pivot point or point of rotation (L1) is 1.45 µm, the length of the hair shaft above the pivot point (L2) is 20.21 µm, and the length ratio of the hair shaft below the pivot point to above the pivot point is about 0.072. Mechanical deformation of the dendrite sheath took place, which was displaced by about 4 nm. The maximum generating electric potential in the microtubules is 1.19 µV. Generated electric potentials are different in each microtubule’s core. It was found that the microtubules near the hair base have higher electric potential than those away from the hair base connection, which has a lower potential in its core. It also presents that the mechanoreception mechanism transformed the transversal deformation of the dendritic sheath into a longitudinal compression of microtubules, the electric potential generated in its core. These electric signals are transferred to the axon and brain of the insect.

Detailed knowledge of the ultrastructure and the stimulation mechanism of trichoid sensilla helps to develop future models of electro-mechanical sensors for other fields of science and to develop an accurate artificial bioinspired sensor design.

## Figures and Tables

**Figure 1 bioengineering-10-00097-f001:**
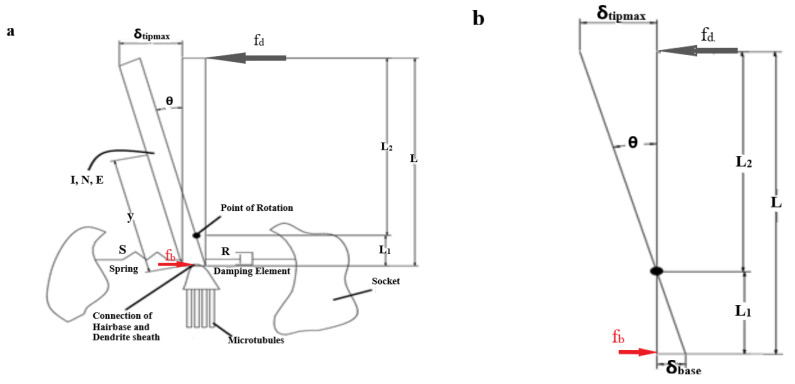
Inverted Pendulum Model. (**a**) Sensilla model as an inverted pendulum. (**b**) The mechanical schematic of sensilla. Θ is the deflection angle, f_d_ is the point load applied at a distal end, fb is the force developed at the hair base due to the distal deflection on the hair shaft, the torsional restoring element S represented by a spring, the torsional damping element R by a dashpot, I is the moment of inertia, N is the torque applied, L is the length of the hair shaft, L_1_ is the length of the hair shaft below the pivot point, L_2_ is the length of the hair shaft above the pivot point, δ_tipmax_ is the distal deflection, and δ_base_ is the basal deflection.

**Figure 2 bioengineering-10-00097-f002:**
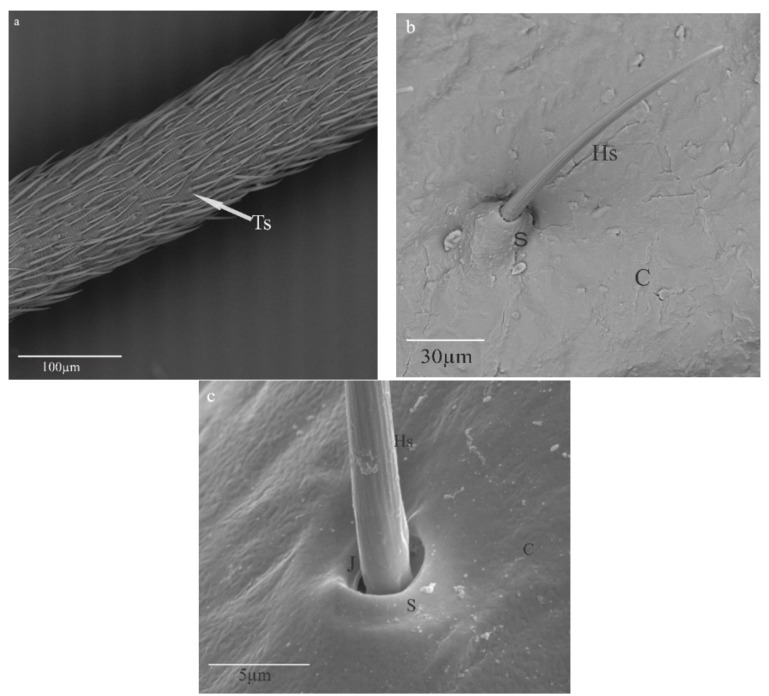
*Nabis rugosus* trichoid sensilla (**a**) Distribution of sensilla over the flagellum segment. (**b**) Single-trichoid sensilla have a hair shaft embedded within a socket on a cuticle C of the flagellum segment. (**c**) The hair shaft is connected to the socket via a joint membrane. Hs: hair shaft, S: socket, C: cuticle, J: joint membrane.

**Figure 3 bioengineering-10-00097-f003:**
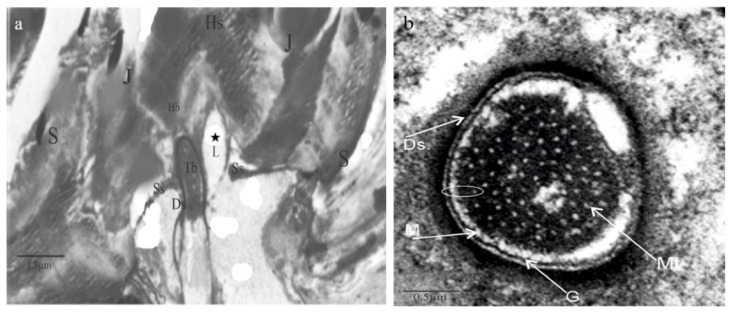
Transmission electron micrographs of the ultra-microstructure of trichoid sensilla in *Nabis rugosus*. (**a**) Sensing plane with the base of the Ts, with the connection of the dendrite sheath to the hair base with the dendritic segment surrounded by the lymph space L indicated by *, thecogen, trichogen, and tormogen cells. (**b**) Cross-sections of the trichoid sensilla dendrite with dendrite sheath and microtubules separated by the membrane and granules along with EDM (electron-dense material)—dark area is EDM. Hs: hair shaft, J: joint membrane, Tb: Tubular body, S: socket, Ds: Dendrite sheath, Ss: socket septum, G: granules, M: membrane, Mt: microtubules.

**Figure 4 bioengineering-10-00097-f004:**
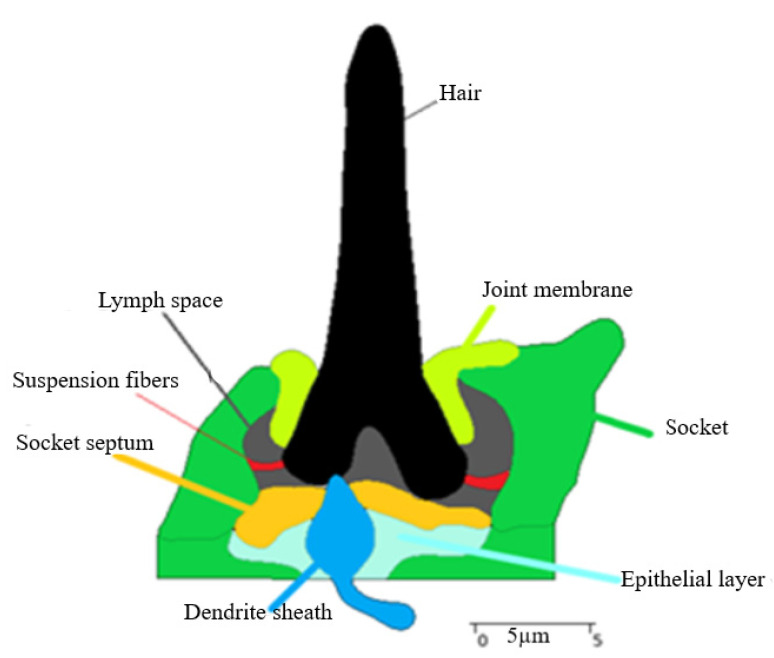
Section view of the trichoid sensilla from AMIRA software consists of the hair, socket, epithelial layer, dendrite sheath, lymph space, socket septum, suspension fibers, and joint membrane.

**Figure 5 bioengineering-10-00097-f005:**
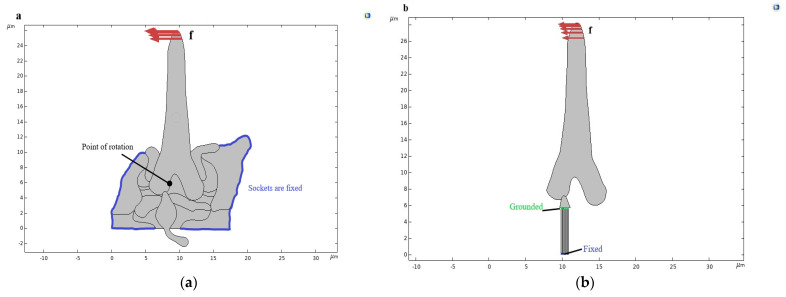
Scheme of the trichoid sensilla model developed from AMIRA sectional view. (**a**) Boundary condition as the socket was fixed (shown in blue color) and external load f_d_ applied on the trichoid sensilla hair shaft. The black point is the point of rotation. (**b**) The scheme of the sensilla with the dendritic sheath and the microtubules, which are fixed at its base and grounded at the dendrite sheath (distal end of the microtubules).

**Figure 6 bioengineering-10-00097-f006:**
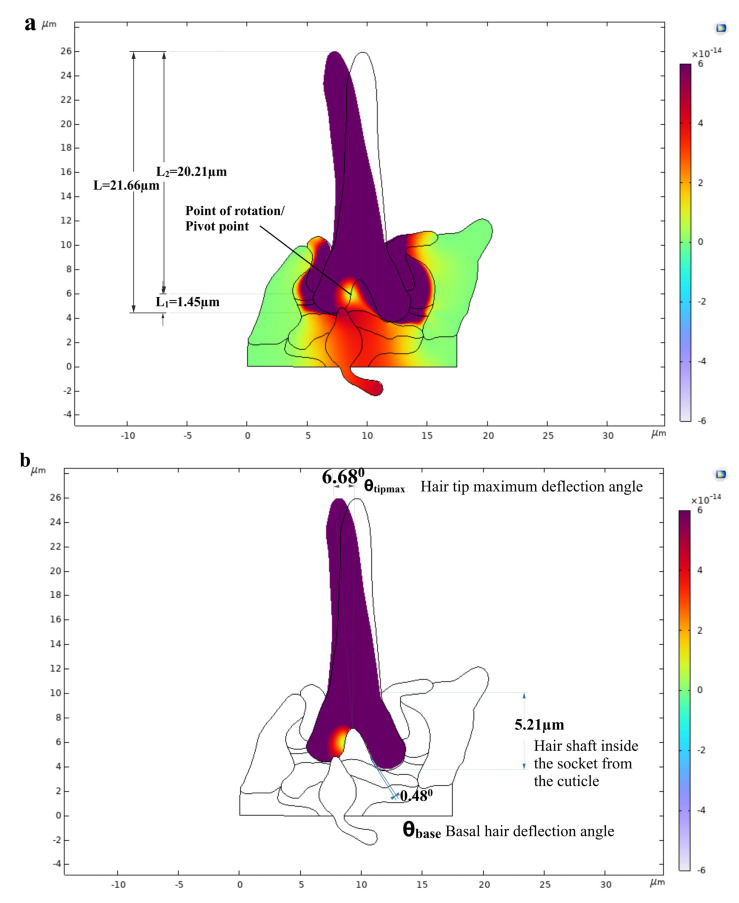
Identification of the parameters (point of rotation, hair-shaft deflection) for the Trichoid sensilla due to applied force on the hair shaft. (**a**) Displacement magnitude, point of rotation near the hair base, and the dendrite connection. (**b**) Deflection angles of hair shaft at the distal tip and the basal part of hair. (**c**) Dendrite sheath displacement field along the horizontal axis. (**d**) Dendrite sheath displacement field along the vertical axis.

**Figure 7 bioengineering-10-00097-f007:**
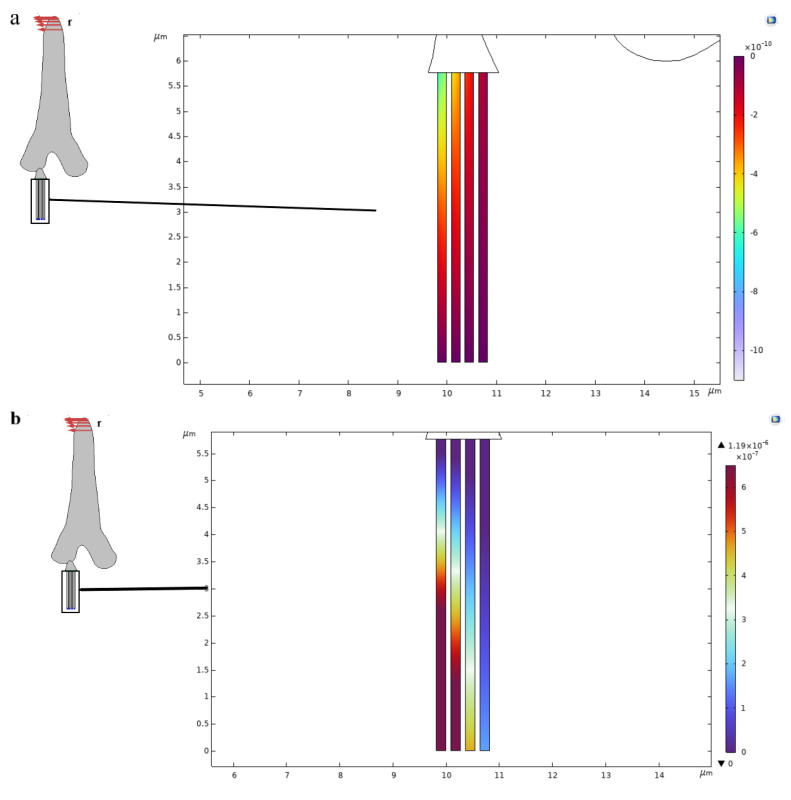
Electro-elastic response of the Tubular body. (**a**) Displacement field of the microtubules along the vertical axis in µm. (**b**) Electrical potential distribution in the microtubules in Volts (V).

**Figure 8 bioengineering-10-00097-f008:**
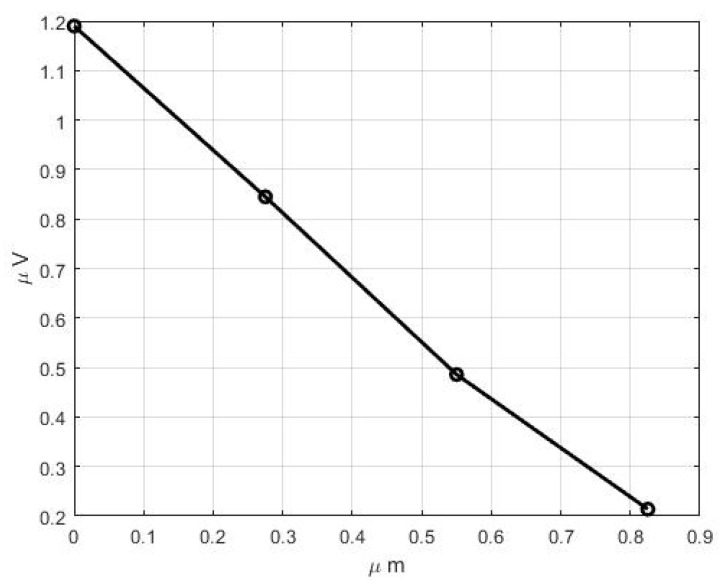
Distribution of electric potential of the individual microtubules at their bases.

**Table 1 bioengineering-10-00097-t001:** Sensilla part mechanical properties.

Part	Youngs Modulus	Poisson’s Ratio	Density	Ref.
Socket and cuticle	2 Gpa	0.3	1100 kg/m^3^	[25]
Hair	1.8 Gpa	0.4	1100 kg/m^3^	[26]
Joint membrane	1.6 Gpa	0.2	1100 kg/m^3^	[27]
Dendrite sheath	1 gpa	0.49	1100 kg/m^3^	[28]
Suspension fibres	500 kpa	0.1	1100 kg/m^3^	[29]
Lymph space	10 Kpa	0.1	1100 kg/m^3^	[30]
Socket septum	0.01 Gpa	0.1	1100 kg/m^3^	[30]

## Data Availability

The data presented in this study are available on request from the corresponding author.

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
