# Peer review of "Ultrastructure of a Mechanoreceptor of the Trichoid Sensilla of the Insect Nabis rugosus: Stimulus-Transmitting and Bio-Sensory Architecture"

_bioengineering, 2023, doi:10.3390/bioengineering10010097_

Round 1

Reviewer 1 Report

Dear Authors,

The manuscript comprehensively presents the ultrastructure morphology of the Nabis rugosus trichoid sensilla using SEM, TEM and the trichoid sensilla two-dimensional model using Amira software.

The manuscript is correct in terms of its content but requires technical corrections. Please pay attention to, in my opinion, too frequent use of capital letters, lack of spaces in many places (e.g. before units) and inappropriate use of italics. It also seems that in some cases, it will be better to give values in nanometers instead of micrometres and resign from excessive accuracy of measurements.

Detailed comments are provided directly in the manuscript.

Best wishes

Author Response

Thank you for the good constructive comments. To reinforce the contents of the manuscript, the frequently used capital letters, the spaces before the units and inappropriate use of the italics are revised as per your suggestions and comments and added to the revised manuscript. All the revised words in the new manuscript are written in Blue. The details of the revision will be explained in the following responses.

Reviewer 2 Report

The manuscript is very well written and should be accepted pending minor revisions. Please see the attachment.

Author Response

Thanks for your comment and suggestions. The changes are made as per your suggestions and have been marked in Purple color in the revised manuscript. The details of the revision will be explained in the following responses.

Reviewer 3 Report

This article reports the ultrastructure morphology of the Nabis rugosus trichoid sensilla using SEM, TEM and the Trichoid sensilla two-dimensional model. The significance of the results is that help to develop miniature tactile bioinspired sensors for analyzing the physical environmental parameters.  This work may help to develop future models of electro-mechanical sensors for other fields of science.

The topic is relevant in the field and interesting for the readers. Largely the work performed is of very good quality. 

Below, there is a list of suggestions that in my opinion would help to improve the manuscript.

1. Material and method are well explained

2. The conclusion is consistent with the evidence and arguments presented and addresses the main question posed.

3. English language spelling and grammar of the manuscript should be checked 

4. Please check the references style

Author Response

Thanks for your good comments. To reinforce the manuscript's contents, English language, spelling, grammar, and references styles are revised per your suggestions and comments and added to the revised manuscript. All the revised references and words in the new manuscript are written in Orange colour. The details of the revision will be explained in the following responses.

Reviewer 4 Report

The present paper thematically is adequate for the Bioengineering journal. The morphological parameters of the trichoid mechnosensilla are well presented, and the model's Mathematical, Geometrical and Numerical Analyses are clearly shown. The actual result of the study is that generated electric potentials are different in each microtubule's core, and the microtubules near the hair base have higher electric potential than those away from the hair base connection. Detailed knowledge of the ultrastructure and the stimulation mechanism of trichoid sensilla are very needed, which helps to develop future models of electro-mechanical sensors for other fields of science and to develop an accurate artificial bioinspired sensor design. Therefore the analysis of the singular different shape/length and the microtubules of the trichoid sensilla in different insects are very useful for biosensory science.The general remarks for the paper:in many lines in the text, some words are written inside the sentences in big letters. It should be changed.

line 222 - in fig 3a, the ' cytoplasmatic reticulum is not visible, and the citation in the result is unnecessary.

Detailed comments are in PDF file.

Author Response

Thanks for your constructive comments and suggestions. The changes are been made as per your suggestions and have been marked in green color in the revised manuscript.
